# Reliability of multi-site UK Biobank MRI brain phenotypes for the assessment of neuropsychiatric complications of SARS-CoV-2 infection: The COVID-CNS travelling heads study

Eugene Duff[1,2,3]*, Fernando Zelaya[4], Fidel Alfaro Almagro[1], Karla L. Miller[1], Naomi Martin[5], Thomas E. Nichols[1,6], Bernd Taschler[1], Ludovica Griffanti[1,7], Christoph Arthofer[1], Gwenaëlle Douaud[1], Chaoyue Wang[1], Thomas W. Okell[1], Richard A. I. Bethlehem[8], Klaus Eickel[9], Matthias Günther[9,10,11], David K. Menon[12], Guy Williams[13], Bethany Facer[14], David J. Lythgoe[4], Flavio Dell'Acqua[4,15,16], Greta K. Wood[17], Steven C. R. Williams[4], Gavin Houston[18], Simon S. Keller[14], Catherine Holden[5], Monika Hartmann[5], Lily George[5], Gerome Breen[5], Benedict D. Michael[17], Peter Jezzard[1], Stephen M. Smith[1], Edward T. Bullmore[8,13]*, on behalf of the COVID-CNS Consortium[¶]

1 Wellcome Centre for Integrative Neuroimaging (WIN FMRIB), University of Oxford, Oxford, United Kingdom, 2 Department of Paediatrics, University of Oxford, Oxford, United Kingdom, 3 Department of Brain Sciences, UK Dementia Research Institute, Imperial College London, London, United Kingdom, 4 Department of Neuroimaging, Institute of Psychiatry, Psychology and Neuroscience, King's College London, London, United Kingdom, 5 Social, Genetic and Developmental Psychiatry Centre, Institute of Psychiatry, Psychology and Neuroscience, King's College London, London, United Kingdom, 6 Big Data Institute, Li Ka Shing Centre for Health Information and Discovery, Nuffield Department of Population Health, University of Oxford, Oxford, United Kingdom, 7 Wellcome Centre for Integrative Neuroimaging, Oxford Centre for Human Brain Activity, Department of Psychiatry, University of Oxford, Oxford, United Kingdom, 8 Department of Psychiatry, University of Cambridge, Cambridge, United Kingdom, 9 mediri GmbH, Heidelberg, Germany, 10 University of Bremen, Bremen, Germany, 11 Fraunhofer MEVIS, Bremen, Germany, 12 Division of Anaesthesia, University of Cambridge, Cambridge, United Kingdom, 13 Wolfson Brain Imaging Centre, Department of Clinical Neurosciences, University of Cambridge, Cambridge, United Kingdom, 14 Department of Pharmacology and Therapeutics, Institute of Systems, Molecular and Integrative Biology, University of Liverpool, Liverpool, United Kingdom, 15 NatBrainLab, Department of Forensics and Neurodevelopmental Sciences, Institute of Psychiatry, Psychology and Neuroscience, King's College London, London, United Kingdom, 16 Sackler Institute for Translational Neurodevelopment, Institute of Psychiatry Psychology and Neuroscience, King's College London, United Kingdom, 17 Clinical Infection Microbiology and Immunology, Institute of Infection, Veterinary and Ecological Sciences, Liverpool, United Kingdom, 18 GE Healthcare, Global Research Organisation, United Kingdom

¶ Membership of the COVID-CNS Consortium is provided in S2 File.
* eduff@imperial.ac.uk (ED); etb23@cam.ac.uk (ETB)

**Data Availability Statement:** Individual de-identified participant data is available through a

## Abstract

### Introduction

Magnetic resonance imaging (MRI) of the brain could be a key diagnostic and research tool for understanding the neuropsychiatric complications of COVID-19. For maximum impact, multi-modal MRI protocols will be needed to measure the effects of SARS-CoV-2 infection on the brain by diverse potentially pathogenic mechanisms, and with high reliability across multiple sites and scanner manufacturers. Here we describe the development of such a

secure online platform (https://cnscovid.wbic.cam.
ac.uk/) which will also host patient data from the
COVID-CNS study. The main UK Biobank brain
MRI analysis pipeline, which has been extended
specifically for this study, is available at https://
www.fmrib.ox.ac.uk/ukbiobank/fbp/. Additional
custom code for COVID-related IDPs is available
from https://www.fmrib.ox.ac.uk/ukbiobank/covid/.
Further resources relating to the COVID-CNS study
are available at covidcns.org.

**Funding:** This research is supported by the COVID-
19 Clinical Neuroscience Study (COVID-CNS), a
Medical Research Council/UK Research and
Innovation (www.ukri.org) funded grant: MR/
V03605X/1 (covidcns.org). Data acquisition at
Cambridge was additionally supported by the
National Institute of Health Research (NIHR)
(www.nihr.ac.uk), Cambridge Biomedical Research
Centre (Mental Health theme) (cambridgebrc.nihr.
ac.uk/research/mental-health) and by
Addenbrooke's Charitable Trust (www.
act4addenbrookes.org.uk). Data acquisition at
King's College London was supported by the NIHR/
Wellcome Trust King's Clinical Research Facility
and NIHR Maudsley Biomedical Research Centre at
South London, Maudsley NHS Foundation Trust,
and King's College London. Data acquisition and
analysis at the University of Oxford was funded in
part by the Wellcome Trust (https://wellcome.org/
grant-funding) [203139/Z/16/Z, 202788/Z/16/Z and
215573/Z/19/Z]. BDM is funded by MR/V03605X/
1, and is supported for additional neurological
inflammation research due to viral infection by
grants from the MRC/UKRI (MR/V007181//1),
MRC (MR/T028750/1) and Wellcome
(ISSF201902/3). ETB is supported by a NIHR
Senior Investigator award. TO is supported by a Sir
Henry Dale Fellowship jointly funded by the
Wellcome Trust (wellcome.org) and the Royal
Society (royalsociety.org) (Grant Number 220204/
Z/20/Z). BDM is supported to conduct COVID-19
neuroscience research by the UKRI/MRC (MR/
V03605X/1). LG, PJ and ED are supported by the
NIHR Oxford Health Biomedical Research Centre
(BRC) (oxfordhealthbrc.nihr.ac.uk). LG is
supported by an Alzheimer's Association Grant
(AARF-21-846366) and by the NIHR Oxford Health
Biomedical Research Centre. KLM is funded by the
Welcome Trust (202788/Z/16/Z). The funders had
no role in study design, data collection and
analysis, decision to publish, or preparation of the
manuscript.

**Competing interests:** EB serves on the scientific
advisory board of Sosei Heptares and as a
consultant for Boehringer Ingelheim,
GlaxoSmithKline, and Monument Therapeutics.

protocol, based upon the UK Biobank, and its validation with a travelling heads study. A
multi-modal brain MRI protocol comprising sequences for T1-weighted MRI, T2-FLAIR, dif-
fusion MRI (dMRI), resting-state functional MRI (fMRI), susceptibility-weighted imaging
(swMRI), and arterial spin labelling (ASL), was defined in close approximation to prior UK
Biobank (UKB) and C-MORE protocols for Siemens 3T systems. We iteratively defined a
comparable set of sequences for General Electric (GE) 3T systems. To assess multi-site
feasibility and between-site variability of this protocol, N = 8 healthy participants were each
scanned at 4 UK sites: 3 using Siemens PRISMA scanners (Cambridge, Liverpool, Oxford)
and 1 using a GE scanner (King's College London). Over 2,000 Imaging Derived Pheno-
types (IDPs), measuring both data quality and regional image properties of interest, were
automatically estimated by customised UKB image processing pipelines (S2 File). Compo-
nents of variance and intra-class correlations (ICCs) were estimated for each IDP by linear
mixed effects models and benchmarked by comparison to repeated measurements of the
same IDPs from UKB participants. Intra-class correlations for many IDPs indicated good-to-
excellent between-site reliability. Considering only data from the Siemens sites, between-
site reliability generally matched the high levels of test-retest reliability of the same IDPs esti-
mated in repeated, within-site, within-subject scans from UK Biobank. Inclusion of the GE
site resulted in good-to-excellent reliability for many IDPs, although there were significant
between-site differences in mean and scaling, and reduced ICCs, for some classes of IDP,
especially T1 contrast and some dMRI-derived measures. We also identified high reliability
of quantitative susceptibility mapping (QSM) IDPs derived from swMRI images, multi-net-
work ICA-based IDPs from resting-state fMRI, and olfactory bulb structure IDPs from T1,
T2-FLAIR and dMRI data.

## Conclusion

These results give confidence that large, multi-site MRI datasets can be collected reliably at
different sites across the diverse range of MRI modalities and IDPs that could be mechanis-
tically informative in COVID brain research. We discuss limitations of the study and strate-
gies for further harmonisation of data collected from sites using scanners supplied by
different manufacturers. These acquisition and analysis protocols are now in use for MRI
assessments of post-COVID patients (N = 700) as part of the ongoing COVID-CNS study.

## Introduction

It is increasingly clear that systemic infection with severe acute respiratory syndrome corona-
virus 2 (SARS-CoV-2) is often associated with acute neurological complications at the time of
infection, as well as post-acute neurological, cognitive and mental health sequelae that can per-
sist for at least 6 months after infection [1]. It seems likely that SARS-CoV-2 infection can
have adverse effects on healthy brain function and structure that account for its broad spec-
trum of neuropsychiatric complications. The causal or pathogenic mechanisms are not yet
defined but are likely to be several, including at least (i) viral infection of the central nervous
system (CNS), (ii) host immune response to infection, and (iii) cerebrovascular disruption.
For precisely targeted interventions, it will be important to know which pathogenic mecha-
nisms are most relevant for which individual patients, or for which syndromically typical
groups of patients [2, 3].

This does not alter our adherence to PLOS ONE policies on sharing data and materials.

Magnetic resonance imaging (MRI) could be a key diagnostic tool in understanding the impacts of systemic SARS-CoV2 infection on the brain and advancing to better treatments for neuropsychiatric complications of COVID-19 in future. Large-scale post-COVID MRI databases will be important because of the geographic, demographic and clinical heterogeneity of neurological, mental health and cognitive syndromes that have been reported as acute or post-acute outcomes of SARS-CoV-2 infection. To acquire such databases requires multi-modal acquisition protocols and analysis pipelines that can be reliably implemented across a variety of scanner manufacturers and models. Ideally, multi-modal MRI protocols for post-COVID research should also be well matched to existing large-scale neuroimaging databases with relevant demographic profiles, such as the UK Biobank database of adults with mean age of 50 years [4]. Here we describe the technical development and validation by a "travelling heads" study of a multi-site protocol for the COVID-CNS consortium, which aims to collect data on ~700 post-COVID neurological cases and controls from a national network of UK sites. In addition to its immediate value for COVID research, this work more generally aims to extend and validate the UK Biobank protocol for a wider range of MRI modalities, and for implementation across multi-site networks using MRI systems provided by more than one major manufacturer, which is expected to be useful for future large-scale MRI studies of non-COVID-related neuropsychiatric disorders.

We started from the principle that a standard brain MRI protocol, robust enough to be reliably implemented across multiple sites and scanners, should also be inclusive of different modalities of MRI that can provide complementary insights into candidate pathogenic mechanisms. For example, the C-MORE consortium for multi-organ MRI studies of post-hospitalised COVID cases [3] has used a set of 7 brain MRI sequences (Table 1), to measure T1-weighted MRI, T2-FLAIR, diffusion MRI (dMRI), susceptibility-weighted MRI (swMRI), and arterial spin labelling (ASL). The inclusion of each of these sequences was justified by their diagnostic relevance to distinct pathogenic mechanisms: e.g., swMRI is a marker of iron deposition and micro-haemorrhages, and ASL measures parameters of regional cerebral blood flow, so both are relevant to vascular mechanisms; T2-FLAIR is a widely used measure of inflammation-related changes in white matter; T1- and dMRI-derived brain structural phenotypes have been found to be associated with immune cell counts in blood samples from post-COVID patients [5]. T1-weighted data have also been used to measure volume and tissue contrast of the olfactory bulb and brain stem structures that may be neurotropically infected via olfactory nerve terminals and other specialist sensory receptors [6, 7]. Thus, the inclusion of sequences in the C-MORE neuro-MRI protocol was well motivated, but the C-MORE requirement to complete all neuroimaging sequences in less than 20 mins, as part of a 70 min multi-organ MRI protocol, meant that some potentially informative sequences were excluded (rfMRI) and others were greatly abbreviated (dMRI, ASL).

In this context, we designed a multi-modal MRI protocol specifically for neuroimaging of post-COVID cases. To optimise comparability with data collected by UKB and C-MORE protocols, we selected Siemens 3T sequences that were as close as possible to these standards, including a multiband sequence for resting state fMRI (implemented in UKB but not in C-MORE) and increasing the scanning time for dMRI and ASL sequences to improve data quality (and biophysical information content) compared to C-MORE. We also defined a set of General Electric (GE) 3T sequences that approximated the parameters of the Siemens sequences (Table 1). Based on our clinical experience to date [2], we rationed the total scanning time of all sequences combined to 30 mins, expecting this to require less than 40 mins of in-scanner time for patients to complete.

To assess the multi-site feasibility and between-site reliability of these protocols, we conducted a "travelling heads" experiment [8] whereby N = 8 healthy volunteers were scanned

**Table 1. Multimodal MRI protocols for COVID-related neuroimaging with Siemens and GE 3T scanners.**

| Modality | Manufacturer | Acquisition Time (min: sec) | Resolution (mm) | Matrix | Key Parameters | UKB Protocol Match | C-MORE Protocol Match |
|---|---|---|---|---|---|---|---|
| T1 (MPRAGE) | Siemens | 4:54 | 1.0x1.0x1.0 | 256x256x208 | TI/TR = 800/2000 ms, R = 2 | Exact | Exact |
| | GE | 4:42 | 1.0x1.0x1.0 | 256x256x208 | TI/TR = 800/2000 ms, R = 2 | | |
| T2 FLAIR (SPACE) | Siemens | 4:32 | 1.0x1.0x1.05 | 256x256x192 | TI/TR = 1800/5000 ms, R = 3 | Similar | Exact |
| | GE | 5:58 | 1.0x1.0x1.0 | 256x256x196 | TI/TR = 1472/5000 ms, R = 2 | | |
| dMRI | Siemens | 7:08 | 2.0x2.0x2.0 | 104x104x72 | TR = 3600 ms, 50 dirs/shell, b = 0, 1000 2000 s/mm2, MB 3 blip-reversed b = 0. AP phase encoding, 36s of PA encoding b = 0 for EPI distortion correction | Exact | Superset |
| | GE | 6:29 | 2.0x2.0x2.0 | 104x104x72 | TR = 3600 ms, 50 dirs/shell, b = 0, 1000 2000 s/mm2, MB 3 blip-reversed b = 0. AP phase encoding, 36s of PA encoding b = 0 for EPI distortion correction | | |
| swMRI | Siemens | 2:08 | 0.9x0.9x3.0 | 256x232x48 | TE1/TE2/TR = 9.4/20/27 ms, R = 2 | Lower resolution | Exact |
| | GE | 2:04 | 0.9x0.9x3.0 | 256x256x48 | TE1/TE2/TE3//TR = 4.9/14.1/23.3/29.5 ms, R = 2 | | |
| ASL segmented 3D-GRASE multi inversion-time PCASL (Siemens only) | Siemens | 3:06 | 3.4x3.4x4.5 | 64x64x32 | TR = variable with PLD, tag = 1800ms, PLDs = 400:400:2000ms, 2 segments, 1 M0 calibration image | Exact. ASL protocol has been added to UKB for post-COVID-19 scanning | Similar |
| ASL (single inversion-time segmented) | Siemens | 5:52 | 1.88*1.88*4.0 interp. from 3.75*3.75*4.0 | 128x128x36 interpolated from 64x64x36 | 3D-GRASE PCASL TR = 4330ms, tag = 1800ms, PLD = 2025ms, 4 reps, 1 M0 calibration image | Not included | Not included |
| | GE | 5:52 | 1.88*1.88*4.0 interp. from 3.75*3.75*4.0 | 128x128x36 interp. from 64x64x36 | FSE stack-of-spirals TR = 4840ms, tag = 1800ms, PLD = 2025ms, 4 reps, 1 M0 calibration image | | |
| Resting fMRI | Siemens | 7:00 | 2.4x2.4x2.4 | 88x88x64 | TE/TR = 39/735 ms, α = 52˚, MB = 8 | Exact | Not Included |
| | GE | 7:21 | 2.4x2.4x2.4 | 88x88x64 | TE/TR = 39/735 ms, α = 52˚, MB = 8 | | |
| Total scanning time | Siemens | 32:33 | | | | | |
| | GE | 33:38 | | | | | |

MPRAGE = Magnetization Prepared RApid Gradient Echo; FLAIR = Fluid-Attenuated Inversion Recovery; SPACE = Sampling Perfection with Application optimized Contrasts using different flip angle Evolution; ASL = Arterial Spin Labeling; PCASL = pseudo-continuous ASL; TR = repetition time; TE = echo time; TI = inversion time; R = in-plane acceleration factor; MB = multi-band acceleration factor; α = flip angle.

once at each of 4 UK sites: 3 using Siemens Prisma 3T systems (Cambridge, Liverpool and Oxford) and 1 using a GE MR750 Discovery 3T system (King's College London). Multi-site consistency of neuroimaging data was evaluated on several dimensions, including quality control (QC) criteria, tissue contrast metrics, and multiple classes of imaging-derived phenotypes (IDPs) estimated using customised UKB image-processing pipelines. Linear mixed effects models were used to estimate components of variance and intra-class correlation coefficients as measures of between-site reliability for each metric and IDP. We focus specifically on two questions of interest: (i) How does between-site and between-manufacturer reliability of multi-modal IDPs estimated from these data compare to the benchmark of test-retest reliability of IDPs estimated from repeated scans of UKB participants using a Siemens Skyra system? (ii) Which are the most (and least) reliable of the thousands of IDPs that can be measured in these data?

## Materials and methods

### Study design and sample

The "travelling heads" design followed previous studies for evaluation of multi-site MRI protocols [3]. Each of N = 8 healthy participants (7F, age range 21–37 y) was scanned 4 times, once at each of the 4 pilot sites: the Wolfson Brain Imaging Centre at the University of Cambridge; the Wellcome Trust-National Institute of Health Research Clinical Research Facility at King's College Hospital, King's College London (KCL); the Liverpool Magnetic Resonance Imaging Centre (LiMRIC) at the University of Liverpool; and the Wellcome Centre for Integrative Neuroimaging at the University of Oxford. Participant age, sex, height and weight statistics are presented in S1 Table.

Due to lockdown restrictions prevailing in the UK at the time of scanning (Dec 2020 –Feb 2021), all participants were recruited at one site (KCL) and the ordering and timing of safe travel to other sites was decided pragmatically. Participants were paid an honorarium to compensate for the time taken to complete the protocol. All participants gave informed consent in writing and the study was approved by the Human Biology Research Ethics Committee, University of Cambridge (HBREC.2020.44). 8 participants were recruited and all completed sessions at each of the sites.

### Scanners and scanning sequences

The Cambridge, Liverpool and Oxford sites all used 3T Siemens MAGNETOM Prisma MRI systems (Siemens Healthineers, Erlangen, Germany) fitted with a 32 channel, receive-only head coil. KCL used a 3T General Electric MR 750 Discovery MRI scanner (GE Healthcare, Waukesha, Wisconsin, USA) fitted with a 32-channel, receive-only head coil (Nova Medical, Wilmington, Massachusetts, USA).

The 3 Siemens scanners implemented the set of 8 sequences summarised in Table 1. The sequence for T1-weighting was implemented identically across UKB, C-MORE and COVID-CNS protocols. dMRI and fMRI were implemented in COVID-CNS exactly as in the UKB protocol (the C-MORE protocol included a shorter dMRI sequence and did not include rfMRI). T2 FLAIR and swMRI sequences were slightly modified from UKB standards in order to more closely match corresponding sequences in the C-MORE protocol. A multi-post label delay (PLD) 3D-GRASE ASL sequence [9] was used that was identical to the ASL sequence used by the UKB COVID study [6] but different to the 2D multi-slice sequence used in C-MORE; a single delay ASL sequence was additionally used to match the ASL pulse sequence of the GE scanner.

The GE scanner implemented an analogous set of 8 sequences (Table 1). In most cases it was possible to approximate the parameters of the Siemens sequences using standard GE sequences. A modified sequence was implemented for SWI. The GE scanner could not implement a multi-post label delay ASL sequence with sufficient similarity to the Siemens implementation, so an additional sequence with a single post label delay was deployed.

### Image processing pipelines and IDPs

Each MRI modality was analysed using custom pipelines for image pre-processing and estimation of multiple MRI contrast metrics and imaging-derived phenotypes (IDPs), derived from the UKB analysis pipelines (www.fmrib.ox.ac.uk/ukbiobank/) [10] and software tools from the FMRIB Software Library [11] and FreeSurfer [12], with DICOM conversion carried out using DCM2NIIX [13]. Pipeline customisations were implemented to accommodate minor differences in imaging parameters between UKB and COVID-CNS protocols, to analyse MRI

modalities not included in the UKB MRI protocol, e.g., ASL, and to analyse MRI data acquired using the GE scanner at KCL. Where protocols matched exactly, analysis pipelines were identical to those used in the C-MORE COVID study [5, 14]. Summaries of pre-processing and IDP estimation are provided below for individual modalities, with further details available elsewhere [5, 6]. For presentation, IDPs reflecting the same phenotypic properties were grouped together into IDP classes [6, 15].

**T1-weighted and T2-FLAIR.** Processing of T1-weighted and T2-FLAIR data included removal of the face, brain extraction, and registration to the MNI152 brain template (Jenkinson 2002, Andersson 2008). We measured spatial signal-to-noise ratio (SNR) and grey/white contrast-to-noise ratio (CNR) as quality control (QC) metrics. As the T1-weighted image was the primary modality for inter-subject registrations, we also measured QC metrics of registration quality. For Siemens scanners we applied post-scan 3D gradient distortion correction as used by the UK Biobank and Human Connectome Project [10]. Implementing the off-scanner 3D gradient distortion correction for Siemens data was not possible for data acquired on the GE scanner. For the sake of consistency of data pre-processing pipelines across scanner manufacturers, a standard GE gradient distortion correction method was implemented for all data. Field map correction was performed using field maps derived from B0 images with FAST was used to segment images into grey matter, white matter, and cerebro-spinal fluid (Zhang 2001). SIENAX [16] was used to estimate volume measures from these segmentations. Grey matter volumes were estimated for each of 139 regions of interest (ROIs) defined by the Harvard-Oxford cortical and subcortical atlases [17] and the Diedrichsen cerebellar atlas [18]. Subcortical volumes were estimated utilizing population priors on shape and intensity variation across subjects [19]. Using an additional non-linear registration procedure, regional volumes of the olfactory bulbs were estimated using T1-weighted, T2-FLAIR and dMRI data, and a parcellation template derived from over 700 UKB individuals [5, 20, 21].

T2-FLAIR pre-processing was very similar to the T1w pipeline (with the T1-weighted image used for registration to the MNI standard template). Images were segmented using BIANCA to identify white matter (WM) hyperintensities (WMH) [22], using the UKB BIANCA training file. Periventricular WMH (pWMH) and deep WMH (dWMH) volumes were defined for complementary subsets of total WM hyperintensities that were, respectively, less than (or more than) 10 mm distant from the lateral ventricles [5].

T1-weighted and T2-FLAIR images were combined in FreeSurfer to model the cortical surface [12, 17]. This analysis produced IDPs encompassing metrics of subcortical segmentation, regional surface area, volume and mean cortical thickness from a number of different parcellations, and grey-white intensity contrasts (expressed as the fractional contrast between white and grey matter intensities as sampled either side of the grey-white cortical boundary) [23]. In total 1448 IDPs were estimated from T1w and T2_FLAIR scans (S1 Fig).

**swMRI.** For the Siemens sequence, the magnitude images from the two echoes of the swMRI data were processed to provide a mapping of $T2^*$ signal decay times [10]. For the GE sequence, swMRI data were acquired using a 3-echo protocol and thus required slightly adjusted post-processing. Key changes to QSM processing were that the coil-combination of phase data was performed on the scanner, and the field perturbation map was estimated using a (magnitude-weighted) linear least-square fit of phase data from all 3 echoes. $T2^*$ mapping was performed using a least square fit of 3-echo magnitude data.

Median $T2^*$ was calculated for 14 subcortical structures defined by registration with the parcellated T1 data [10]. To enable qualitative neurological assessment of individual patients, the median phase and magnitude data were processed to provide maps highlighting features indicative of abnormal iron deposition, e.g., due to microbleeds. Quantitative susceptibility mapping (QSM) was also performed using the phase data, following a recently developed UKB

QSM pipeline [24]. Susceptibility maps were generated using the iLSQR algorithm [25], with susceptibility values reported relative to the susceptibility of CSF. In total 28 IDPs were measured from swMRI scans.

**ASL.** For the Siemens sequence, we used the BASIL tools in FSL to estimate maps of cerebral blood flow (CBF) from single-PLD data and CBF and arterial transit time (ATT) from multi-PLD data [26]. BASIL analysis included motion correction and distortion correction using a fieldmap derived from the blip up/down dMRI data. Label and control images were subtracted and a kinetic model was fitted with modelling of the macrovascular component [27]. The M0 calibration image acquired without ASL preparation or background suppression was used to quantify CBF in the CSF for calibration. Tissue-specific CBF was achieved by projecting grey and white partial volume maps from the T1w image segmented by FAST into the ASL native space. Grey and white matter masks were defined using partial volume thresholds of 50% and 80%, respectively. To avoid dependence on site-specific T1w data, T1w grey matter masks derived from each scan site were normalised into MNI space to identify voxels present in all masks for the estimation of mean grey matter CBF and ATT. In total 4 IDPs were measured from both the multi- and single-PLD ASL data.

**fMRI.** For the Siemens sequence, the multiband-8 fMRI data were corrected for gradient and EPI distortions, motion-corrected using linear alignment using the UKB Resting fMRI pipeline [10], and aligned to the T1w image via a single-band reference image. For the GE sequence, the first high-contrast fMRI image prior to magnetisation stabilisation was used for T1w registration. FIX ICA-based denoising was applied using the UKB training dataset [28]. FIX identifies and removes artefacts related to measured head-motion, cardio-vascular cycles, acquisition variability and other phenomena. Two sets of resting-state networks derived from group ICA decompositions of UKB reference data (25 and 100 component decompositions with 21 and 55 neural components respectively) were projected onto the pre-processed resting state fMRI data in a dual-regression analysis [29]. Whole brain functional connectivity matrices were compiled from full and partial correlations (210 and 1485 elements for 25 and 100 component decompositions respectively). The amplitudes (standard deviations) of spontaneous activity at each regional node were also estimated (21 and 55 elements) [10]. As individual connections showed low test-retest reliability in the UKB dataset, we used a dimension-reduction approach which applied ICA to all functional connectivity IDPs to produce 6 primary modes of variation [15]. These six modes were projected onto the individual's functional connectivity matrix and used as additional IDPs. Finally, four fMRI QC IDPs were defined (alignment discrepancy, head motion, initial and FIX cleaned tSNR). Excluding the individual connectivity nodes, 86 IDPs were assessed from the fMRI data.

**dMRI.** For the Siemens and GE sequences, dMRI data were closely matched to the UKB sequence and processed using UKB pipelines with minimal alterations [10]. The AP-encoding data were pre-processed to remove effects of eddy currents, head motion, and slice dropouts, followed by gradient distortion correction. DTIFIT used the b = 1000 shell for diffusion tensor image fitting [30] to estimate parameters including fractional anisotropy (FA), tensor mode (MO) and mean diffusivity (MD). The multi-shell data were processed with NODDI (Neurite Orientation Dispersion and Density Imaging) [31], to produce microstructural parameters including ICVF (intra-cellular volume fraction, an index of white matter neurite density), ISOVF (isotropic or free water volume fraction), L1, L2 and L3 eigenvalues of the diffusion tensor, and ODI (orientation dispersion index; a measure of within-voxel tract disorganisation). These parameters were summarised using two approaches: first, using a white-matter tract skeleton analysis producing average values for 48 standard-space tract masks [32]; and second, using probabilistic tractography: BEDPOSTx was used to model multi-fibre tract orientation structure and PROBTRACKx for probabilistic tractography with crossing fibre

modelling [10]. For 27 major tracts defined in AutoPtx (http://fsl.fmrib.ox.ac.uk/fsl/fslwiki/ AutoPtx), the posterior mean fractional voxel occupancy defined the weights for weighted-mean estimates of each DTI/NODDI parameter [10]. In total 677 IDPs were measured from the dMRI data, including 2 QC IDPs.

## Statistical analysis and UK Biobank benchmarking

Site and scanner manufacturer can affect the distribution of phenotypes derived from brain images, adding variability and reducing experimental power in multi-site studies. Site effects limited to phenotype value shifts and scale changes are easily modeled if they can be estimated, and will result in within-site subject-type ranking being preserved across sites. Here we characterise the effects of site on the mean value and scale of IDPs, and compare intra-class correlations (ICCs) of IDPs measured 4 times for each subject scanned at 4 different sites in the travelling heads study, against ICCs of the same IDPs measured twice for each subject (with a 2y interval) at the same site as part of the longitudinal data previously acquired as part of of the UKB imaging enhancement programme [33]. Biobank-style deconfounding was not possible here due to the low subject numbers repeated-measures design. Additionally, translation of Biobank deconfounding estimates was not possible as our pilot cohort differs considerably from the demographic profile of the UKB cohort. However, these deconfounding procedures will be implemented in the analyses of the principal study data from the COVID-CNS consortium.

Site effects were assessed using repeated-measures ANOVAs, testing the null hypotheses of zero between-site difference in mean IDPs. Normality was assessed with Shapiro-Wilk, and sphericity was assessed using Mauchly's test. A lack of sphericity indicates a violation of ANOVA assumptions, an independent effect of site on IDP measures, i.e., between-site. differences in variation of an IDP. A Greenhouse-Geisser correction for sphericity was applied if sphericity was detected. These tests were made for each of 2243 (total) IDPs (excluding IDPs representing individual functional network connections). A power analysis indicated that the available subject numbers would permit the reliable identification of "large" site effects (eta-squared = 0.2) with power = 0.87, while smaller site effects (eta-squared = 0.1) would be less reliably detected (power = 0.51). When assessing different patterns of effects across different classes of IDPs, we used the false discovery rate (FDR = 5%, within each class of IDP), to control type 1 errors. Site-specific effects on each IDP were estimated twice: once using all the analysable data (from 4 sites, including 1 GE site), and once using only Siemens data (from 3 sites). This allowed us to investigate site-differences in IDP location or sphericity that were likely related to between-manufacturer differences in MRI scanners.

Intra-class correlation coefficients (ICCs) were estimated for pairs of IDP vectors (N = 8), each vector comprising measurements of the same IDP in the same subjects at one of 4 possible scanning sites [34, 35]. The ICC provides a measure of reliability by quantifying the within-subject similarity of each outcome metric or IDP across different sites. ICCs were estimated by linear mixed effects modeling of variance components, accounting for between-subject and between-site variance, using the lme4 package in R (Bates et al., 2015) [36]. We report ICCs estimated by modelling site as a fixed effect ("consistent" ICC, or ICC(3,1)). We estimated ICCs twice: once using all analysable data from 4 sites, including 1 GE site; and once using only Siemens data from 3 sites. ICC values between 0.5 and 0.8 are generally considered to indicate fair to good reliability, and ICCs greater than 0.8 or 0.9 are indicative of good or very good reliability [37].

To benchmark the between-subject and between-site reliability of each IDP measured using the COVID-CNS protocol, we compared these ICCs from the travelling heads study to

comparable ICCs estimated in the UKB cohort. In this design, N = 2,817 largely healthy middle-aged participants were each scanned twice (with mean between-scan interval = 2.25 y; SD = 0.12) at the same one of 4 possible sites, all using the same scanner for multi-modal MRI (Siemens Skyra 3T). As noted, the MRI sequences for COVID-CNS were based on similar or identical sequences for T1, T2 FLAIR, dMRI, swMRI and fMRI used in the UKB enhancement cohort (Table 1). Hence, we could directly compare test-retest and between-site consistency of IDPs measured in the UKB and COVID-CNS cohorts. We estimated ICCs between the test and retest IDP measurement vectors for N = 8 participants, repeatedly, randomly sampled from the total UKB dataset (N = 2,817; 1000 random samples). This allowed us to estimate the distribution of ICCs for repeated MRI measurements on N = 8 participants using data with no site or scanner contributions to variance. From these resampled distributions, we computed P-values for a two-sided test of the null hypothesis that the ICCs estimated from the travelling heads experimental data were sampled from the same distribution as the ICCs estimated from repeated measurements on comparable sub-samples of the UKB cohort.

## Results

### Sample

Eight participants (7 F; mean age = 23.5 y; SD = 5.8) were successfully scanned at all four sites, with between-site intervals ranging from 1–14 days.

### T1w and T2-FLAIR images

Quality control of T1w and T2 FLAIR images disclosed no deviations in quality of registration (Fig 1a) across sites or with UK Biobank. T1w SNR and CNR measures from Siemens sites were consistent with the UKB population distributions. However, the GE scanner produced images with higher measures of inverse SNR and CNR (equivalent to lower SNR/CNR) than other sites for all subjects ($P<0.05$) (Fig 1b). For Siemens sites, across structural IDPs, there was negligible evidence for site-dependent variation in IDP mean values or scaling, and normality violations were rare. ICC distributions matched those observed in UK Biobank. Overall, across structural and other modalities, ICC values were correlated across UK Biobank and travelling heads samples. Across all IDPs, r = 0.43 for Siemens scanners only and r = 0.39 when including the GE site. With a large proportion of IDPs having consistently high ICCs

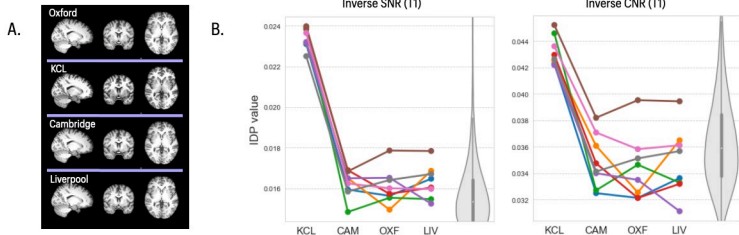

**Fig 1. T1 images, inverse SNR and inverse CNR metrics across four sites. A)** Representative T1 images of the same subject scanned at each of 4 sites in the travelling heads study. **B)** *left panel*, plots of inverse signal-to-noise ratio (iSNR) for 8 subjects (coloured lines) scanned at each of 4 sites (x-axis labels); *right panel*, plots of inverse contrast-to-noise ratio (iCNR) for the same subjects and sites. The grey violin plots in both panels indicate the equivalent distributions of T1 iSNR and iCNR, respectively, in the UK Biobank reference dataset, using matched random sampling of N = 8 participants. Box and whiskers represent inter-quartile range and 95% confidence intervals respectively. The iSNR and iCNR metrics are comparable across Siemens sites (CAM = Cambridge, OXF = Oxford, LIV = Liverpool) and aligned with the UKB benchmark distribution. Both iSNR and iCNR are higher for the GE site (KCL = Kings College London) ($P < 0.05$), indicating lower SNR and CNR.

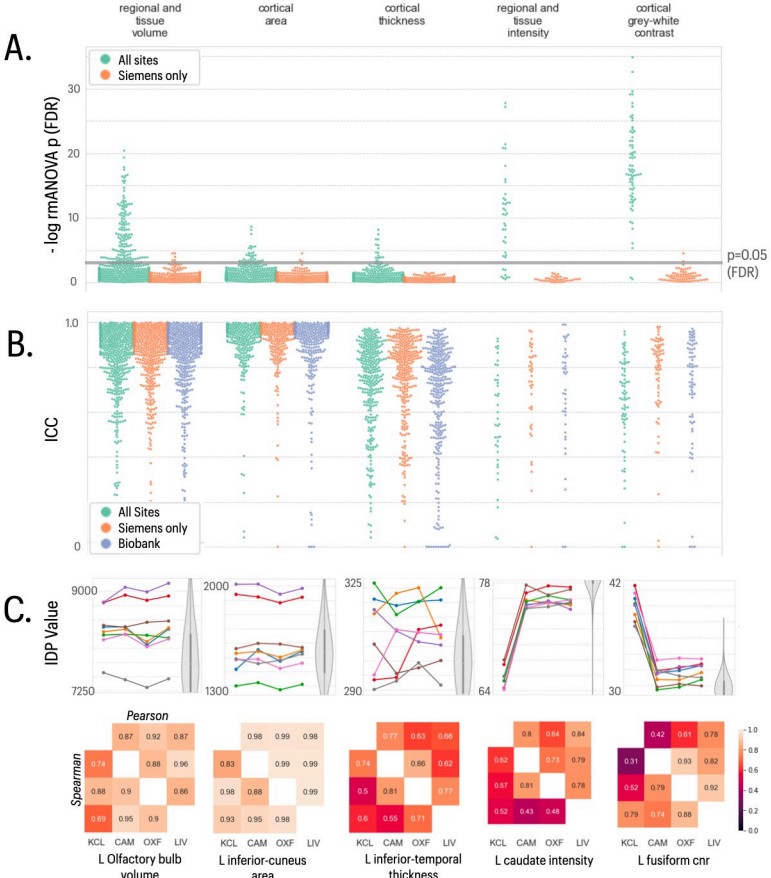

**Fig 2. Statistical results for five classes of structural MRI-derived phenotypes.** In the top two panels, each column represents results for a different class of IDP, from left to right: regional and tissue volumes, cortical area, cortical thickness, regional and tissue intensity, and cortical grey-white contrast. **A)** Distribution of log-transformed *P*-values from repeated measures ANOVA testing for a site effect on the mean value of individual IDPs in each class; the solid horizontal line represents the *P*-value equivalent to FDR = 5%. Green dots represent IDPs fitted to the ANOVA model including data from all four sites; orange dots represent *P*-values for each IDP fitted to the ANOVA including only data from the three Siemens sites (Cambridge, Oxford, Liverpool). There are more significant between-site differences in mean IDPs, across all 5 classes, when the GE data from KCL are included in the analysis **B)** Swarm plots showing distribution of intra-class correlation coefficients (ICCs) for the same IDPs, estimated for each pair of all 4 sites (green points), for each pair of the three Siemens sites (orange points), and for comparable test-retest data drawn from the UKB cohort (blue points). Between-site reliability was generally high for all IDP classes compared to the UKB benchmark, whether or not GE data were included in the analysis. **C)** Each column represents finer-grained results for representative IDPs from each class of IDP: from left to right, left olfactory bulb volume, left precuneus area, left inferior temporal cortical thickness, left caudate intensity and left fusiform CNR. *Top row*, plots of each IDP for 8 subjects (coloured lines) scanned at each of 4 sites (x-axis labels); the grey violin plots indicate the distributions of the corresponding IDP in the UK Biobank reference datase, using matched random sampling of N = 8 participants. Box and whiskers represent inter-quartile range and 95% confidence intervals respectively. *Bottom row*, correlations between each pair of sites for each IDP: upper triangle, Pearson's correlations; lower triangle, Spearman's correlations.

(e.g. >0.7), these correlations are estimated from a relatively low level of variation in ICCs across IDPs (see S1 Fig). To confirm our statistical results given the limited sample size, we repeated our analysis using the parametric Friedman's test. This produced very similar distributions of P-values across IDPs (S2 Fig).

Morphometric IDPs, measuring regional volumes and surface areas, showed limited evidence of site-dependent variations in their mean values for Siemens scanners (repeated measures ANOVA; FDR = 5%, Fig 2a), and no evidence of significant between-site differences in

scaling (Mauchy's test for sphericity, $P > 0.05$). The GE scanner site had an impact on between-site differences in mean value for a subset of these IDPs. However, consistency across all sites, measured by ICCs, was generally very good for these IDPs (mean ICC >0.9) and did not differ from ICC measures of test-retest consistency in the UKB dataset (Fig 2b). Similar results were observed for regional cortical thickness IDPs derived from T1w and T2-FLAIR data. There was some regional variability in between-site (and test-retest) reliability of cortical thickness, but ICCs were typically indicative of good to very good reliability (mean ICC ~ 0.8), matching those observed in UK Biobank.

Tissue intensity and grey-white contrast IDPs were again consistent across Siemens sites, but for many IDPs showed significant differences at the GE site. There were significant between-site differences in mean tissue intensity and grey-white contrast for 79% and 97%, respectively, of regional IDPs (RM ANOVA, FDR = 5%). Sphericity tests also pointed to altered variance at the GE site for a proportion of tissue intensity (23%) and grey-white contrast (34%) IDPs. Grey-white contrast measured on the GE data was generally lower than in the Siemens data, reflecting the effects observed in the global SNR and CNR measures (Fig 1b). Between-site reliability for these IDPs across the 3 sites using Siemens scanners was slightly higher (mean ICC = 0.69, SD = 0.24) than between-site reliability across all 4 sites (mean ICC = 0.61 SD = 0.23), compared to a UK Biobank mean ICC of 0.66 (SD = 0.17).

White matter hyper-intensity volumes (WMHs) derived from T2-FLAIR images of the healthy young adults scanned in the travelling heads study were typically low, as expected in this age range (21–37 y). However, there were significant mean differences between sites in both deep and periventricular WMH volumes (RM ANOVA; FDR = 5%), due to greater WMH volumes in the GE data, with correspondingly lower levels of between-site reliability (Fig 3). There were no significant mean differences between Siemens sites in deep or periventricular WMH volumes and between-site reliability for the 3 Siemens sites was very good (ICC = 0.95, SD = 0.01), comparable to test-rest reliability in the UKB data (ICC = 0.90,

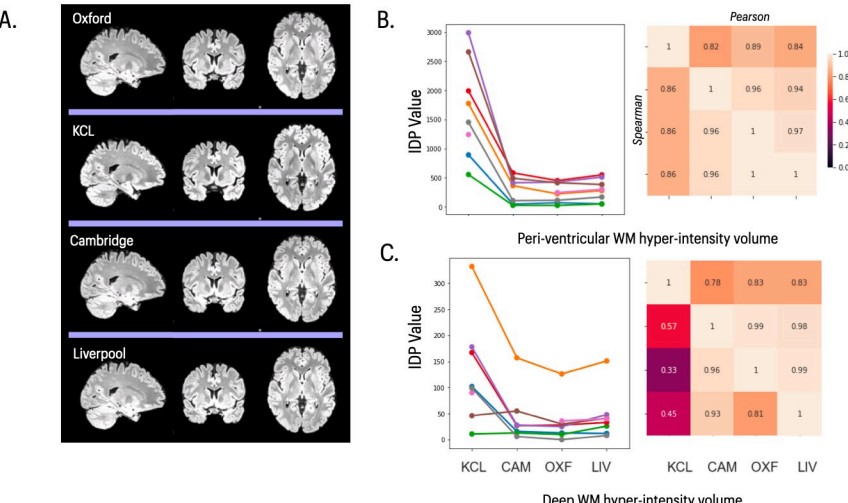

**Fig 3. T2 FLAIR images and statistical results for T2-derived IDPs. A)** Representative T2 FLAIR images of the same subject scanned at each of 4 sites in the travelling heads study. **B)** *left panel*, peri-ventricular white matter hyperintensity volume for 8 subjects (coloured lines) scanned at each of 4 sites (x-axis labels); *right panel*, correlations between each pair of sites. **C)** *left panel*, deep white matter hyperintensity volume for 8 subjects (coloured lines) scanned at each of 4 sites (x-axis labels); *right panel*, correlations between each pair of sites. In both **B)** and **C)**, the upper triangle of the matrix shows Pearson's correlations and the lower triangle shows Spearman's correlations; and both IDPs were estimated using BIANCA.

SD = 0.06), and greater than between-site reliability over all 4 sites in the travelling heads data (ICC = 0.51, SD = 0.12). These findings are somewhat unsurprising given that the software tool for WMH measurement (BIANCA) was trained on data collected from the Siemens MRI protocol. When adequate training data are available from the GE protocol, and in older subjects where higher WMH volumes are expected, it will be important to retrain the BIANCA algorithm on both Siemens and GE data, and this may improve consistency of WMH IDPs across scanners from the different manufacturers [38].

### Susceptibility weighted imaging

We assessed regional estimates of $T2^*$ signal decay and quantitative estimates of susceptibility (QSM) derived from the swMRI images. There was limited evidence of site-specific variation in IDP means or scaling (Fig 4a). Estimates of regional $T2^*$ had poor between-site reliability across all 4 sites in the travelling heads data (mean ICC = 0.34, SD = 0.24) (Fig 4b). QSM-derived IDPs had generally better between-site reliability (all sites: ICC = 0.67, SD = 0.13; Siemens only sites: ICC = 0.76, SD = 0.14), comparable to good ctest-retest reliability in the UKB data (ICC = 0.66). Lower reliability was observed for QSM IDPs measured in smaller subcortical structures (amygdala, nucleus accumbens) in both travelling heads and UKB datasets.

**dMRI.** Diffusion weighted images were successfully acquired and analysed at all sites. Visualisation and basic QC metrics showed consistent image quality across sites. IDPs corresponding to multiple diffusion parameters (FA, MO, MD, ICVF, ISOVF and OD) were estimated regionally for each of multiple white matter tracts. As for other modalities, some IDPs showed evidence for site-specific differences in means, driven by a difference between the Siemens sites and single GE site (Fig 5a). Some WM tract ICVF (23%) and ISOVF (20%) diffusion parameter IDPs showed evidence of non-normality. Overall, there was good to very good between-site reliability (mean ICCs > 0.7), matching those observed in the UKB (Fig 5b). The GE site showed limited consistency with other sites for WM tract FA, diffusivity and ISOVF, reducing ICCs for these categories of IDPs.

**fMRI.** Resting fMRI was successfully acquired at all sites. There were no significant between-site differences in mean tSNR (before or after ICA-based artefact removal with FIX), indicating similar levels of signal quality across all sites, with QC metrics commensurate with those observed in the UKB data (Fig 6). As individual functional connectivity (FC) IDPs reflecting pairwise connectivity do not show a high level of reliability (see above), we assessed 6 modes of variation of functional network connectivity shown to be reliable in UKB [15]. We also assessed individual node amplitudes. These IDPs in general did not show site-specific variations in mean (Fig 6A), although two of the six (2,6) showed evidence of sphericity violations. Between-site reliability was fair-to-good for node amplitudes (all 4 sites: mean = 0.36, SD = 0.17; Siemens only sites, mean = 0.55, SD = 0.19), and comparable to the UKB (mean = 0.48, SD = 0.27). The 6 RSN connectivity modes showed very good reliability, with mean ICC = 0.67 (SD = 0.18) for all sites and ICC = 0.75 (SD = 0.25) for Siemens sites, compared to the excellent reliability seen in the UKB (mean ICC = 0.89, SD = 0.11).

**Arterial spin labelling.** For both the single PLD sequence (acquired on all sites) and the multi-PLD sequence (acquired on the three Siemens sites only), we assessed estimates of grey and white matter mean perfusion. Due to acquisition challenges, ASL was not successfully acquired at all sites for every subject (n = 6 at Cambridge), and some variation in data quality was apparent. Estimates of CBF were higher in the single PLD sequence compared to the multi PLD sequence, but there was no evidence of between-site mean differences in estimated perfusion within the same sequence. The between-site reliability for the single PLD sequence was fair (ICC = 0.47), but high when considering Siemens sites only ICC = 0.92. The Siemens-only

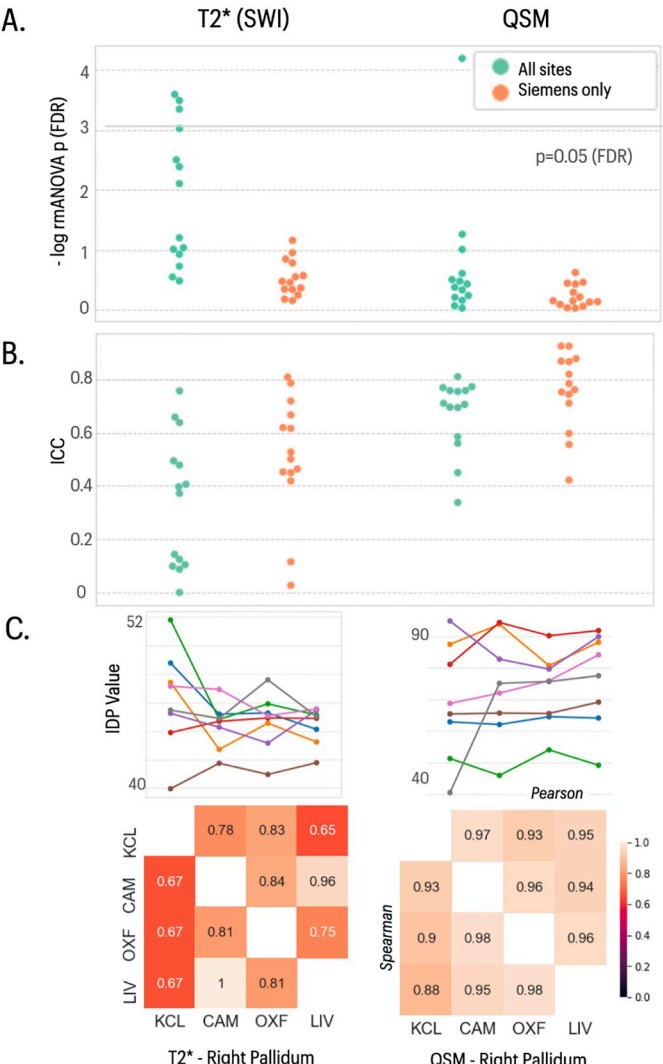

**Fig 4. Statistical results for SWI-derived IDPs.** In the top two panels, the left column shows data for 14 IDPs derived from T2* data and the right column shows data for 14 IDPs derived from QSM data. **A)** Distribution of log-transformed *P*-values from repeated measures ANOVA testing for a site effect on the mean value of individual IDPs in each class; the solid horizontal line represents the P-value equivalent to FDR = 5%. Green dots represent IDPs fitted to the ANOVA model including data from all four sites; orange dots represent *P*-values for each IDP fitted to the ANOVA including only data from the three Siemens sites (Cambridge, Oxford, Liverpool). There were more significant between-site differences in mean IDPs when the GE data from KCL were included in the analysis **B)** Swarm plots showing distribution of intra-class correlation coefficients (ICCs) for the same IDPs, estimated for each pair of all 4 sites (green points), and for each pair of the three Siemens sites (orange points). **C)** Each column represents finer-grained results for representative IDPs from each class of IDP: from left to right, T2* right pallidum, QSM right pallidum. *Top row*, plots of each IDP for 8 subjects (coloured lines) scanned at each of 4 sites (x-axis labels). *Bottom row*, correlations between each pair of sites for each IDP: upper triangle, Pearson's correlations; lower triangle, Spearman's correlations.

multi PLD sequence had fair reliability, largely driven by a discrepancy in one subject across Oxford and Liverpool acquisitions (ICC = 0.53) (Fig 7).

## Discussion

This study provides a survey of the multi-site, multi-manufacturer reliability of hundreds of distinct multi-modal neuroimaging measures. For the COVID-CNS project, it provides insights

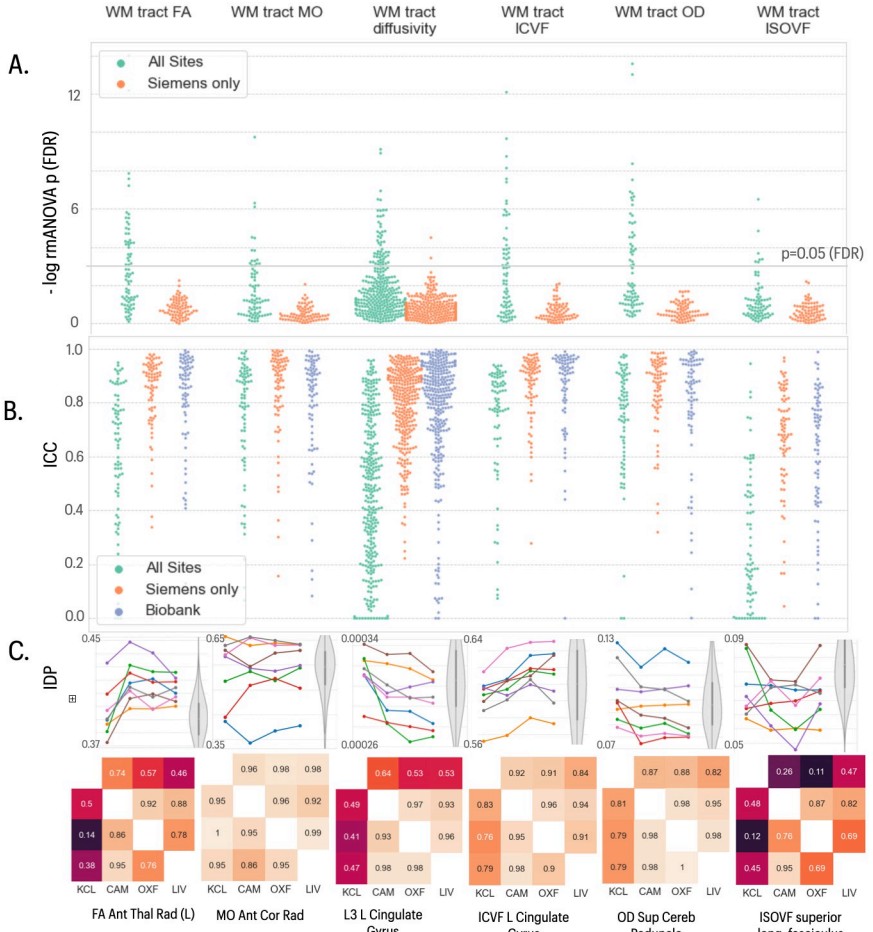

**Fig 5. Statistical results for five classes of dMRI-derived phenotypes.** In the top two panels, each column represents results for a different class of IDP, from left to right: white matter (WM) tract FA, WM tract MO, WM tract diffusivity, WM tract ICVF, WM tract OD and WM tract ISOVF. **A)** Distribution of log-transformed *P*-values from repeated measures ANOVA testing for a site effect on the mean value of individual IDPs in each class; the solid horizontal line represents the *P*-value equivalent to FDR = 5%. Green dots represent IDPs fitted to the ANOVA model including data from all four sites; orange dots represent *P*-values for each IDP fitted to the ANOVA including only data from the three Siemens sites (Cambridge, Oxford, Liverpool). There were more significant between-site differences in mean IDPs, across all 5 classes, when the GE data from KCL were included in the analysis. **B)** Swarm plots showing distribution of intra-class correlation coefficients (ICCs) for the same IDPs, estimated for each pair of all 4 sites (green points), for each pair of the three Siemens sites (orange points) and for comparable test-retest data drawn from the UKB cohort (blue points). Between-site reliability was generally high for all IDP classes compared to the UKB benchmark when only Siemens sites were included in the analysis. **C)** Each column represents finer-grained results for representative IDPs from each class of IDP: from left to right, FA right anterior thalamic radiation. MO left corona radiata, L3 left cingulate gyrus, ICVF left cingulate gyrus, OD superior cerebellar peduncle, and ISOVF superior longitudinal fasciculus. *Top row*, plots of each IDP for 8 subjects (coloured lines) scanned at each of 4 sites (x-axis labels); the grey violin plots indicate the distributions of the corresponding IDP in the UK Biobank reference dataset, using matched random sampling of N = 8 participants. Box and whiskers represent inter-quartile range and 95% confidence intervals respectively. *Bottom row*, correlations between each pair of sites for each IDP: upper triangle, Pearson's correlations; lower triangle, Spearman's correlations.

that can guide the design of harmonised MRI protocols for assessment of brain changes caused by multiple putative pathogenic mechanisms of neuropsychiatric complications of SARS-CoV2 infection. More generally, the study is of relevance to the expanding number of clinical research studies utilising multi-modal imaging (acquisition and analysis) protocols derived from UK Biobank, including a number of additional studies focused on the neurological impact of COVID-

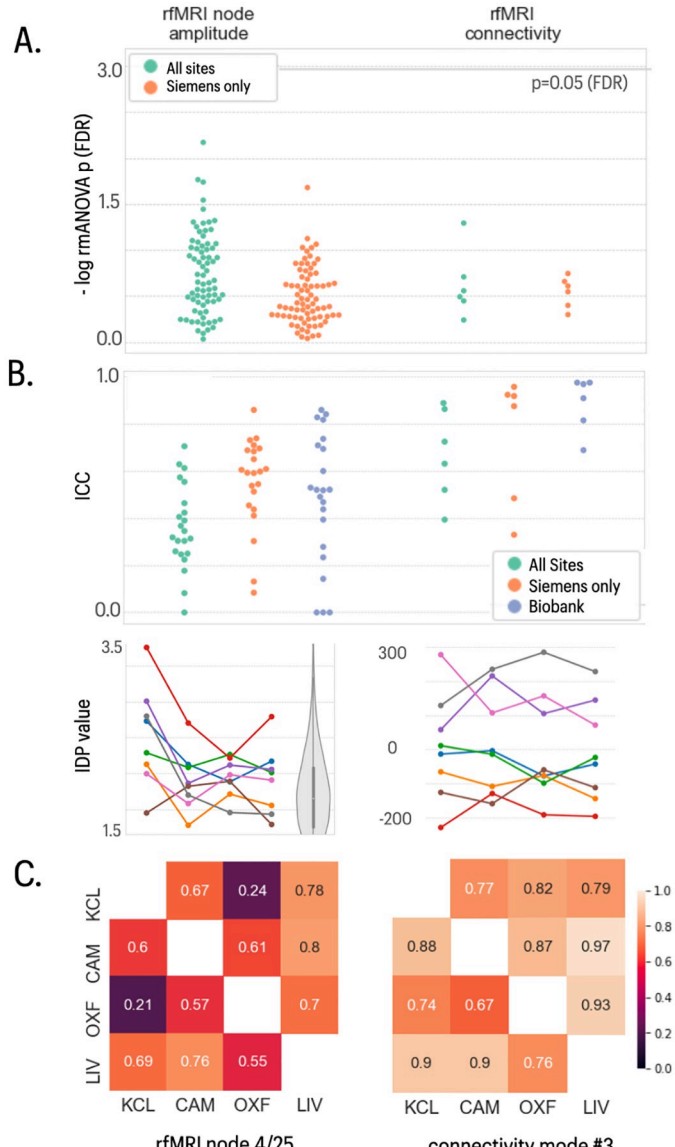

**Fig 6. fMRI data quality and IDP summaries.** The two columns show data on fMRI node amplitude and fMRI connectivity IDPs. Both represent IDPs derived from 25- and 100-node ICA-based parcellations. The fMRI connectivity IDPs represent 6 modes of variation across the functional connectivity network matrices derived from both parcellations. **A**) Distribution of log-transformed *P*-values from repeated measures ANOVA testing for a site effect on the mean value of individual IDPs in each class; the solid horizontal line represents the *P*-value equivalent to FDR = 5%. Green dots represent IDPs fitted to the ANOVA model including data from all four sites; orange dots represent *P*-values for each IDP fitted to the ANOVA including only data from the three Siemens sites (Cambridge, Oxford, Liverpool). **B**) Swarm plots showing distribution of intra-class correlation coefficients (ICCs) for the same IDPs, estimated for each pair of all 4 sites (green points), for each pair of the three Siemens sites (orange points) and for comparable test-retest data drawn from the UKB cohort (blue points). Between-site reliability was generally high for all IDP classes compared to the UKB benchmark, whether or not GE data were included in the analysis. **C**) Each column represents finer-grained results for representative IDPs from each class of IDP: from left to right, fMRI node 4/ 25 (medial visual RSN) and summary connectivity mode #3 [15]. *Top row*, plots of each IDP for 8 subjects (coloured lines) scanned at each of 4 sites (x-axis labels); the grey violin plot indicates the distribution of the corresponding IDP in the UK Biobank reference dataset. *Bottom row*, correlations between each pair of sites for each IDP: upper triangle, Pearson's correlations; lower triangle, Spearman's correlations.

A. **Single PLD:** 2025 ms

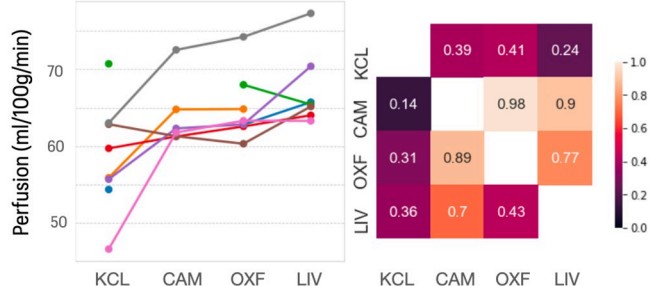

B. **Multi PLD:** = 400,800,1200,1600,2000ms (Siemens)

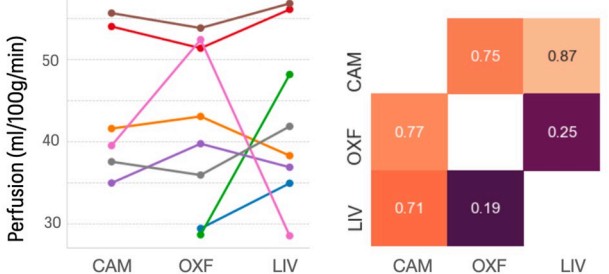

**Fig 7. ASL data IDP summaries. A)** Grey matter mean CBF perfusion (ml/100g/min) measurements for the single post-label delay (PLD) sequence used across all sites. **B)** Grey matter mean CBF perfusion measurements for the multi-PLD sequence available only on the Siemens sites. Raw data is plotted to the left; the cross-site correlation matrices to the right (upper triangle, Pearson's correlation; lower triangle, Spearman's correlation).

19 [6, 14]. Overall, our results demonstrate generally good to very good (ICCs 0.8–0.9) levels of between-site reliability of imaging derived phenotypes estimated across a wide range of brain MRI modalities in data collected from 4 UK sites. In particular, the 3 sites using Siemens Prisma platforms had levels of between-site reliability that were comparable to the high levels of test-retest reliability estimated from repeated measures on participants sampled from the UKB database. When data were included from the site using a scanner supplied by a different manufacturer (GE), certain IDP classes had lower levels of between-site reliability, but between-site reliability remained acceptable for most IDPs. These results give confidence that large, multi-site studies can be used to expand the cohort sizes of neuroimaging studies for clinical research studies of COVID and other pathogenically heterogeneous disorders.

Variability in IDPs across sites may be induced by variation in the contrast obtained by specific sequences and scanner setups or technical variation in signal levels, scaling, or SNR. Travelling heads studies provide a powerful means by which to detect site-specific variations in these features in advance of multi-site population studies. In a healthy-participant travelling heads study, ICC depends on intrinsic inter-subject variation in the travelling heads cohort to drive measures of reliability. As such, ICC may be an imperfect measure to compare IDPs, as between-subject variability may not reflect the observed effect size in the condition of interest for individual IDPs (e.g. neurological effects of COVID). Nevertheless, ICCs are valuable when it is expected that clinical effect sizes will be on the approximate scale of individual variation, and for comparison to other datasets (e.g. UKB). While N = 8 provides limited statistical power for the identification of subtle differences across sites in individual IDPs, here it was able to provide an overall pattern of results indicating that there will not be substantial loss of statistical power when introducing new sites.

## Reliability of multi-modal, multi-site MRI measurements

The between-site reliability for the 3 sites using Siemens Prisma scanners allow us to evaluate which MRI sequences and IDPs were most (and least) operationally and statistically reliable under the best-case scenario of nearly identical scanners at multiple sites. The most reliably collected MRI sequences were T1w, T2-FLAIR and dMRI; the least reliably collected MRI sequence was ASL (N> = 6). This is perhaps unsurprising given the relative novelty of these ASL sequences, which are well-established for research at specialised centres but had not previously been used for large-scale clinical studies at all sites participating in the travelling heads study. Further, ASL is likely to be particularly sensitive to intra- and inter-data physiological variability affecting blood flow. With larger subject numbers, exploration of measures of relative levels of regional blood flow may identify more stable IDPs.

The most reliably estimated IDPs were geometric grey matter phenotypes (cortical volume, surface area, thickness), and white matter microstructural phenotypes (FA, ODI etc). Between-site reliability for these two classes of IDPs was excellent in the travelling heads data and comparable to the ceiling level of test-retest reliability of the same classes of IDP in the UKB dataset. Less reliably estimated IDPs were typically derived from the less reliably collected ASL data; but all other classes of IDP had good-to-very good levels of both between-site and test-rest reliability. It was notable that the ICCs for between-site and test-retest reliability were positively correlated across all IDPs derived from Siemens data in the travelling heads and UKB studies, indicating that some IDPs are inherently more robust to both between-site and within-subject sources of variation. This may have implications for the power to detect case-control differences in clinical studies using this set of multi-modal MRI sequences. For example, assuming comparable effect sizes and sample sizes across modalities, T1w, T2-FLAIR and dMRI-derived IDPs will clearly have greater power to detect case-control differences by virtue of their lower (between-site) variability.

## Between-manufacturer reliability of multi-modal MRI measurements

The GE platform increased between-site variability for many classes of IDP, showing significant differences in mean and reductions in ICC. Clearly, this increased between-site variability was driven by differences in MRI sequences and data between Siemens and GE scanner platforms. Despite careful preparatory alignment of the GE sequences to approximate as closely as possible the parameters of the Siemens sequences, there were some irreducible differences between Siemens and GE protocols due to the hardware constraints of differently manufactured scanners. This affected tissue contrast metrics, like grey/white matter contrast, which differed between Siemens and GE scanners, whereas geometric grey matter IDPs were generally more robust.

The reliability of white matter hyperintensity volume estimation was notably poor when GE data were included in the analysis, but this may be at least partly attributable to the fact that WMH volumes were estimated in healthy young adults (not usually expected to have large amounts of WMHs) and was using a software tool that had been trained on Siemens-only data. Further training of WMH segmentation tools on data acquired from GE as well as Siemens platforms in older subjects would likely improve the reliability of this key marker of inflammation-related changes in white matter.

For a nationally-scaled study of post-COVID patients, these data clearly point to a trade-off between increasing recruitment rates (and ultimately sample size) by including sites using scanners supplied by different manufacturers *versus* maximising between-site reliability (and thus reducing spurious sources of variability) by restricting sites to those that are using scanners supplied by the same manufacturer. Geographical differences in the incidence of COVID,

and in operational capacity for research studies under pandemic conditions, motivated formation of a large and nationally representative network of scanning sites. We considered that the generally good-to-very good levels of reliability for most IDPs across all sites in this pilot study were sufficient to support this more inclusive strategy of using sites with either Siemens or GE scanners, with the caveat that this will entail loss of power to detect case-control differences in terms of IDPs derived from ASL and other modalities which were most difficult to harmonise between manufacturers. Between-site offsets in the mean and scaling of IDP values could be corrected statistically post hoc by harmonisation or modelling methods such as COMBAT [39] or Generalised Additive Modelling [40]. These approaches can require good numbers of patients and controls at each site, which may limit multi-site clinical studies to sites that can recruit good numbers of participants.

## Methodological issues

It is a strength of this study that we have assessed reliability across a wide range of MRI modalities and imaging-derived phenotypes, using data collected from different MRI systems and at different sites. It is also a strength that we have been able to benchmark between-site reliability for the majority of IDPs against comparable estimates of test-retest reliability in the UKB data. However, sample size for the travelling heads study was small, meaning that results were potentially vulnerable to the effects of 1 or 2 outlying observations, and confidence intervals were generally wide. We made best efforts, under the pragmatic constraints of urgently responding to a public health crisis, to align GE and Siemens sequences prior to data acquisition. However, we cannot claim that the between-manufacturer reliability results are perfectly optimised or would be unimprovable by future, more intensive work on Siemens-like sequences for sites using scanners supplied by GE or other manufacturers, to align with UKB and C-MORE standards for COVID neuroimaging. The sample for the travelling heads experimental study was considerably younger than the UK Biobank population, and the expected COVID-CNS consortium cohort of post-hospitalisation patients. It comprised 7 females and 1 male, and additional demographic data were not obtained. While we expect the relative youth of the travelling heads sample to have negligible impact on our assessment of between-site reliability of multimodal MRI, this demographic disparity precludes more detailed comparisons between these data and technically comparable data on the UK Biobank and COVID-CNS cohorts. The results also indicate strong prospects for the wider integration of COVID-related clinical neuroimaging data, particularly when sequences are reasonably aligned across studies.

## Conclusion

These results represent a guide to the generally good-to-very good levels of between-site reliability that are immediately attainable for multi-modal MRI across a national network of collaborating sites using different scanner platforms. The UK Biobank multimodal imaging protocols, which we have translated here to other sites and scanner models, present an attractive suite of protocols for new studies to consider adopting to ensure strong reusability of data.

## Supporting information

**S1 Fig.** Scatterplots of Intra-class Correlation Coefficients (ICCs) for IDPs derived from UK Biobank repeat-scan data and Travelling Heads dataset A. Siemens sites (r = 0.43). B. All sites (r = 0.39). Each point corresponds to ICC for an individual IDP.
(TIF)

**S2 Fig. Comparison of ANOVA and Friedman tests.** Comparison of FDR-corrected P-values for site effects derived using repeated measures ANOVA (orange) and non-parametric Friedman (green) tests.
(TIF)

**S1 Table. Participant information.**
(DOCX)

**S1 File. Covid-19 Clinical Neuroscience Study (COVID-CNS) consortium membership and affiliation.**
(DOCX)

**S2 File. List of IDPs.**
(DOCX)

# Acknowledgments

We thank the volunteers who participated in the travelling heads study and the radiography staff who collected data at all four sites. We thank Fraunhofer MEVIS, Bremen, Germany, for provision of the Siemens 3D-GRASE ASL sequence.

# Author Contributions

**Conceptualization:** Eugene Duff, Fernando Zelaya, Karla L. Miller, Christoph Arthofer, Gwenaëlle Douaud, Chaoyue Wang, David K. Menon, Guy Williams, Steven C. R. Williams, Simon S. Keller, Gerome Breen, Benedict D. Michael, Peter Jezzard, Stephen M. Smith, Edward T. Bullmore.

**Data curation:** Eugene Duff, Fernando Zelaya, Fidel Alfaro Almagro, Karla L. Miller, Naomi Martin, Ludovica Griffanti, Richard A. I. Bethlehem, Klaus Eickel, Matthias Günther, David K. Menon, Bethany Facer, David J. Lythgoe, Flavio Dell'Acqua, Greta K. Wood, Gavin Houston, Simon S. Keller, Catherine Holden, Monika Hartmann, Lily George, Gerome Breen, Benedict D. Michael, Peter Jezzard, Edward T. Bullmore.

**Formal analysis:** Eugene Duff, Fidel Alfaro Almagro, Thomas E. Nichols, Edward T. Bullmore.

**Funding acquisition:** David K. Menon, Guy Williams, Steven C. R. Williams, Gerome Breen, Benedict D. Michael, Stephen M. Smith, Edward T. Bullmore.

**Investigation:** Eugene Duff, Fernando Zelaya, Naomi Martin, Bernd Taschler, Ludovica Griffanti, Christoph Arthofer, Chaoyue Wang, Thomas W. Okell, Richard A. I. Bethlehem, Greta K. Wood, Catherine Holden, Gerome Breen, Benedict D. Michael, Peter Jezzard, Edward T. Bullmore.

**Methodology:** Eugene Duff, Fernando Zelaya, Fidel Alfaro Almagro, Karla L. Miller, Thomas E. Nichols, Bernd Taschler, Ludovica Griffanti, Christoph Arthofer, Gwenaëlle Douaud, Chaoyue Wang, Thomas W. Okell, Klaus Eickel, Matthias Günther, David K. Menon, Guy Williams, David J. Lythgoe, Flavio Dell'Acqua, Greta K. Wood, Steven C. R. Williams, Gavin Houston, Simon S. Keller, Gerome Breen, Stephen M. Smith, Edward T. Bullmore.

**Project administration:** Naomi Martin, Guy Williams, Bethany Facer, Greta K. Wood, Steven C. R. Williams, Simon S. Keller, Catherine Holden, Monika Hartmann, Lily George,

Gerome Breen, Benedict D. Michael, Peter Jezzard, Stephen M. Smith, Edward T. Bullmore.

**Resources:** David K. Menon, Guy Williams, Bethany Facer, Flavio Dell'Acqua, Greta K. Wood, Steven C. R. Williams, Simon S. Keller, Gerome Breen, Peter Jezzard, Edward T. Bullmore.

**Software:** Eugene Duff, Fidel Alfaro Almagro, Thomas E. Nichols, Bernd Taschler, Ludovica Griffanti, Christoph Arthofer, Gwenaëlle Douaud, Chaoyue Wang, Thomas W. Okell, Richard A. I. Bethlehem, Peter Jezzard, Stephen M. Smith.

**Supervision:** Eugene Duff, Fernando Zelaya, Thomas E. Nichols, Steven C. R. Williams, Simon S. Keller, Gerome Breen, Benedict D. Michael, Peter Jezzard, Stephen M. Smith, Edward T. Bullmore.

**Validation:** Eugene Duff.

**Visualization:** Eugene Duff.

**Writing – original draft:** Eugene Duff, Fernando Zelaya, Bernd Taschler, Simon S. Keller, Edward T. Bullmore.

**Writing – review & editing:** Eugene Duff, Fernando Zelaya, Fidel Alfaro Almagro, Karla L. Miller, Naomi Martin, Thomas E. Nichols, Bernd Taschler, Ludovica Griffanti, Christoph Arthofer, Gwenaëlle Douaud, Chaoyue Wang, Thomas W. Okell, Richard A. I. Bethlehem, Klaus Eickel, Matthias Günther, David K. Menon, Guy Williams, Bethany Facer, David J. Lythgoe, Flavio Dell'Acqua, Greta K. Wood, Steven C. R. Williams, Gavin Houston, Catherine Holden, Monika Hartmann, Lily George, Gerome Breen, Benedict D. Michael, Peter Jezzard, Stephen M. Smith, Edward T. Bullmore.

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
