## [Decision Letter · Decision Letter 0]

21 Mar 2022

PONE-D-22-02931Reliability of multi-site UK Biobank MRI brain phenotypes for the assessment of neuropsychiatric complications of SARS-CoV-2 infection: the COVID-CNS travelling heads studyPLOS ONE

Dear Dr. Duff,

Thank you for submitting your manuscript to PLOS ONE. After careful consideration, we feel that it has merit but does not fully meet PLOS ONE’s publication criteria as it currently stands. Therefore, we invite you to submit a revised version of the manuscript that addresses the points raised during the review process.

We look forward to receiving your revised manuscript.

Kind regards,

Pew-Thian Yap

Academic Editor

PLOS ONE

Journal Requirements:

" This research is supported by the COVID-19 Clinical Neuroscience Study (COVID-CNS)(covidcns.org), an Medical Research Council/UK Research and Innovation (www.ukri.org) funded grant: MR/V03605X/1. Data acquisition was additionally supported by the National Institute of Health Research (NIHR) (www.nihr.ac.uk), Cambridge Biomedical Research Centre (Mental Health theme) (cambridgebrc.nihr.ac.uk/research/mental-health) and by Addenbrooke’s Charitable Trust (www.act4addenbrookes.org.uk). LG, PJ and ED are supported by the NIHR Oxford Health Biomedical Research Centre (BRC) (oxfordhealthbrc.nihr.ac.uk) ETB is supported by a NIHR Senior Investigator award. TO is Reliability of multi-modal MRI-derived brain phenotypes for multi-site assessment of neuropsychiatric complications of SARS-CoV-2 infection Duff et al., January 2022, submission to PLOS ONE supported by a Sir Henry Dale Fellowship jointly funded by the Wellcome Trust (wellcome.org) and the Royal Society (royalsociety.org) (Grant Number 220204/Z/20/Z). This research was funded in part by the Wellcome Trust [203139/Z/16/Z, 202788/Z/16/Z and 215573/Z/19/Z]. For the purpose of open access, the author has applied a CC-BY public opyright licence to any Author Accepted Manuscript version arising from this submission. The funders had no role in study design, data collection and analysis, decision to publish, or preparation of the manuscript.

We note that you have provided funding information. However, funding information should not appear in the Funding section or other areas of your manuscript. We will only publish funding information present in the Funding Statement section of the online submission form. 

"This work was partly funded by the UKRI COVID-CNS consortium (covidcns.org). Data acquisition was additionally supported by the National Institute of Health Research (NIHR) (www.nihr.ac.uk), Cambridge Biomedical Research Centre (Mental Health theme) (cambridgebrc.nihr.ac.uk/research/mental-health) and by Addenbrooke’s Charitable Trust (www.act4addenbrookes.org.uk). 

LG, PJ and ED are supported by the NIHR Oxford Health Biomedical Research Centre (BRC) (oxfordhealthbrc.nihr.ac.uk). ETB is supported by a NIHR Senior Investigator award. TO is supported by a Sir Henry Dale Fellowship jointly funded by the Wellcome Trust (wellcome.org) and the Royal Society (royalsociety.org) (Grant Number 220204/Z/20/Z).

This research was funded in part by the Wellcome Trust [203139/Z/16/Z, 202788/Z/16/Z and 215573/Z/19/Z]. For the purpose of open access, the author has applied a CC-BY public copyright licence to any Author Accepted Manuscript version arising from this submission. 

This research is supported by the COVID-19 Clinical Neuroscience Study (COVID-CNS), an Medical Research Council/UK Research and Innovation (www.ukri.org) funded grant: MR/V03605X/1. 

"I have read the journal's policy and the authors of this manuscript have the following competing interests: EB serves on the scientific advisory board of Sosei Heptares and as a consultant for Boehringer Ingelheim, GlaxoSmithKline, and Monument Therapeutics."

Reviewers' comments:

Reviewer's Responses to Questions

**Comments to the Author**

1. Is the manuscript technically sound, and do the data support the conclusions?

Reviewer #1: Partly

Reviewer #2: Partly

2. Has the statistical analysis been performed appropriately and rigorously? 

Reviewer #1: No

Reviewer #2: Yes

3. Have the authors made all data underlying the findings in their manuscript fully available?

Reviewer #1: Yes

Reviewer #2: Yes

4. Is the manuscript presented in an intelligible fashion and written in standard English?

Reviewer #1: Yes

Reviewer #2: Yes

5. Review Comments to the Author

Reviewer #1: In this manuscript the reliability of features extracted from multi-modal MRIs were evaluated across different sites and scanners, by scanning 8 participants at 4 UK sites: 3 using Siemens

PRISMA scanners (Cambridge, Liverpool, Oxford) and 1 using a GE scanner (King’s

College London). It gave confidence that large, multi-site MRI datasets can be

collected reliably at different sites across the diverse range of MRI modalities and IDPs

that could be mechanistically informative in COVID brain research. This work is important to give guidelines on how to handle the cross-site/cross-scanner data heterogeneity of various MRI phenotypes.

The major limitation of the paper is the small sample size---only 8 participants were considered. With such a small sample size, the statistical analysis should be more rigid. It seems like I cannot find the confidence intervals in tables or Figures of the ICCs. Have some confounding factors removed from the IDP similar to what UK Biobank did, such as motion parameters from rest fMRI, ICV from brain regional volume etc.? The authors needed to describe in details. The author should better provide more demographic information about the 8 participants, for example, are they British, percentages of males/females? Could the authors show the scatterplots of UKBiobank reliabilities versus your IDP reliabilities for both the cases of all scanners and siemens only? They should be highly correlated if they are correct. As the sample size is too small, the author may use more non-parametric statistical test instead of those test with Gaussian assumptions.

Minor issues are listed in the following.

1) In page 19, for fMRI traits, why there are 3464 IDPs? could the author explain. Similarly for dMRI, the author considered FA, MO, MD, ICVF, ISOVF, ODI, and there may probably be (48+27)*6 but why there are 675 IDPs?

2) to correct some typos. For example, for standard deviation, the author uses "sd" somewhere while uses "SD" in other places.

3) There are four rows in Figure 6. Does the subfigures in row 3 belong to Figure 6 C) or Figure 6 B)? Please make it clear.

Reviewer #2: This paper presents a multi-modal MRI protocol to reliably scan post-COVID patients across different sites. More than 2,000 IDPs were obtained by processing the acquired T1w, T2w, dMRI, rs-fMRI, sw-MRI, and ASL images. Between-site reliability of these IDPs were tested using intra-class correlations (ICC) and compared against the test-retest reliability of the same IDPs in scans from UK Biobank. The results for sites with same scanner manufacturer are more reliable compared to the one with different scanner manufacturer. Thus, reliability of the proposed multi-modal MRI protocol is limited to same type of scanner, which may not guarantee comparable data acquired using different scanner types.

I have following questions:

1. The authors mentioned that 1,073 IDPs were obtained from T1w and T2-FLAIR scans. Can they list few IDPs that were obtained? And what are the metrics of subcortical segmentation?

2. For post-acquisition processing of structural MRIs, why a standardized distortion correction method not applied to scans from Siemens and GE scanners?

3. “To avoid dependence on site-specific T1w data, we used T1w data from all sites to define generic masks for estimation of mean grey matter CBF and ATT.” How are these generic masks obtained? Are multi-site T1w data spatially normalized to a common space to get mask?

4. Are dMRI data collected with single phase encoding?

5. How are the weighted-mean summaries of diffusion parameters for each tract computed?

6. Can the reliability of IDPs be improved using post-acquisition harmonization?

6. PLOS authors have the option to publish the peer review history of their article (what does this mean?). If published, this will include your full peer review and any attached files.

Reviewer #1: No

Reviewer #2: No

---

## [Author Response · Author response to Decision Letter 0]

7 Jun 2022

See response to reviewers document

---

## [Decision Letter · Decision Letter 1]

13 Jul 2022

PONE-D-22-02931R1Reliability of multi-site UK Biobank MRI brain phenotypes for the assessment of neuropsychiatric complications of SARS-CoV-2 infection: the COVID-CNS travelling heads studyPLOS ONE

Dear Dr. Duff,

Thank you for submitting your manuscript to PLOS ONE. After careful consideration, we feel that it has merit but does not fully meet PLOS ONE’s publication criteria as it currently stands. Therefore, we invite you to submit a revised version of the manuscript that addresses the points raised during the review process.

We look forward to receiving your revised manuscript.

Kind regards,

Pew-Thian Yap

Academic Editor

PLOS ONE

Journal Requirements:

Reviewers' comments:

Reviewer's Responses to Questions

**Comments to the Author**

1. If the authors have adequately addressed your comments raised in a previous round of review and you feel that this manuscript is now acceptable for publication, you may indicate that here to bypass the “Comments to the Author” section, enter your conflict of interest statement in the “Confidential to Editor” section, and submit your "Accept" recommendation.

Reviewer #1: All comments have been addressed

Reviewer #2: All comments have been addressed

2. Is the manuscript technically sound, and do the data support the conclusions?

Reviewer #1: Yes

Reviewer #2: Yes

3. Has the statistical analysis been performed appropriately and rigorously? 

Reviewer #1: Yes

Reviewer #2: Yes

4. Have the authors made all data underlying the findings in their manuscript fully available?

Reviewer #1: Yes

Reviewer #2: Yes

5. Is the manuscript presented in an intelligible fashion and written in standard English?

Reviewer #1: Yes

Reviewer #2: Yes

6. Review Comments to the Author

Reviewer #1: The author has fully addressed my comments, and I think the analysis now is rigid. One of my minor concern is the resolution/quality of many of your figures are low, for example, Figure 2~6. I cannot see clearly the axis labels/fonts/numbers in many places of the figures. Please improve the figure resolution and increase those font sizes.

Reviewer #2: (No Response)

7. PLOS authors have the option to publish the peer review history of their article (what does this mean?). If published, this will include your full peer review and any attached files.

Reviewer #1: No

Reviewer #2: No

---

## [Author Response · Author response to Decision Letter 1]

8 Aug 2022

Thank you for your careful assessment by of this submission. We have made the following changes requested to the manuscript:

Review comments to the author reviewer #1:

One of my minor concern is the resolution/quality of many of your figures are low, for example, Figure 2~6. I cannot see clearly the axis labels/fonts/numbers in many places of the figures. Please improve the figure resolution and increase those font sizes.

We have edited figures 2-7 to ensure the clarity of all text within. 

We have also reviewed our references and have updated cited preprints that have now been accepted for publication.

---

## [Editor Report · Decision Letter 2]

12 Aug 2022

Reliability of multi-site UK Biobank MRI brain phenotypes for the assessment of neuropsychiatric complications of SARS-CoV-2 infection: the COVID-CNS travelling heads study

PONE-D-22-02931R2

Dear Dr. Duff,

We’re pleased to inform you that your manuscript has been judged scientifically suitable for publication and will be formally accepted for publication once it meets all outstanding technical requirements.

Kind regards,

Pew-Thian Yap

Academic Editor

PLOS ONE
---

## [Editor Report · Acceptance letter]

8 Sep 2022

PONE-D-22-02931R2 

Reliability of multi-site UK Biobank MRI brain phenotypes for the assessment of neuropsychiatric complications of SARS-CoV-2 infection: the COVID-CNS travelling heads study 

Dear Dr. Duff:

I'm pleased to inform you that your manuscript has been deemed suitable for publication in PLOS ONE. Congratulations! Your manuscript is now with our production department. 

Kind regards, 

on behalf of

Dr. Pew-Thian Yap 

Academic Editor

PLOS ONE